# Is *Orius sauteri* Poppius a Promising Biological Control Agent for Walnut Aphids? An Assessment from the Laboratory to Field

**DOI:** 10.3390/insects12010025

**Published:** 2021-01-04

**Authors:** Ting Wang, Ping Zhang, Chenyang Ma, Muhammad Yasir Ali, Guizhen Gao, Zhaozhi Lu, Myron P. Zalucki

**Affiliations:** 1Laboratory of Insect Ecology and Molecular Biology, College of Plant Health and Medicine of Qingdao Agriculture University, Qingdao 266109, China; liuxiaoxian18@mails.ucas.ac.cn (T.W.); zhaozhi@ms.xjb.ac.cn (M.Y.A.); 2Xinjiang Institute of Ecology and Geography, Chinese Academy of Sciences, Urumqi 830011, China; zhangping16@mails.ucas.ac.cn (P.Z.); machenyang19@mails.ucas.ac.cn (C.M.); 3College of Forestry and Horticulture, Xinjiang Agricultural University, Urumqi 830052, China; sunjing@xjau.edu.cn; 4School of Biological Sciences, The University of Queensland, Brisbane 4072, Australia; m.zalucki@uq.edu.au

**Keywords:** biological control efficiency, *Chromaphis juglandicola* Kaltenbach, dusky-veined aphid, *Panaphis juglandis* Goeze, walnut aphid

## Abstract

**Simple Summary:**

Walnut aphids are major pests of walnut orchards with few commercially available natural enemies except parasitoids. The predatory bug (*Orius sauteri)* was assessed as a potential biological control agent against two walnut aphid species. This bug has a strong capacity for consuming both aphid species. Biocontrol efficacy of *O. sauteri* for each species was high (more than 70%), but declined about 20% when both aphid species present on the same leaf together. This might be due to the competition among two species of aphids tested. Three statistical approaches were employed to show that *O. sauteri* is a promising biocontrol agent. The commercial release of *O. sauteri* should be considered for the coexisting aphids in walnut orchards to promote economic and environmental benefits.

**Abstract:**

Walnut aphids are major pests of walnut production with few commercially available natural enemies. We conducted laboratory and field experiments to evaluate the potential of *Orius sauteri* Poppius (Anthocoridae), a predatory bug, as a biological control agent against two walnut aphid species: the dusky-veined aphid (*Panaphis juglandis* Goeze) and the walnut aphid (*Chromaphis juglandicola* Kaltenbach). Both species co-occur on walnut trees; *P. juglandis* is distributed on the upper surface (adaxial) of leaves while *C. juglandicola* is found on the lower surface (abaxial) of leaves. Based on functional response experiments, *O sauteri* had a strong capacity for consuming both aphid species. Biocontrol efficacy of *O. sauteri* for each species in the laboratory and field experiments was high, 77% for *P. juglandis* and 80% for *C. juglandicola*, regardless if one or two predators being present. However, biocontrol efficacy declined 15–25% for *C. juglandicola* and 20–50% for *P. juglandis* when both aphid species were present on the same leaf. The efficacy of *O. sauteri* under (semi)-field conditions gave similar findings based on the percentage reduction of aphids and change in population growth rates of aphids. The reduced biocontrol efficacy of the predatory bug against mixed species populations of aphids can be explained by competition between the aphid species and differences in their preferred location on leaves. Our experiments showed that *O. sauteri* is a promising biocontrol agent, but biocontrol efficacy may decline when both aphid species are present on walnut trees. This should be considered in the commercial release of *O. sauteri* in walnut orchards to promote economic and environmental benefits of walnuts production.

## 1. Introduction

The biological control of pests by their natural enemies is a crucial ecosystem service that substantially contributes to protecting the environment as well as maintaining biodiversity [1]. Natural enemies are commercially available for the protection of crops and trees [2]. Biological control has been widely employed as a sustainable component of integrated pest management worldwide with economic, environmental, and social benefits [3], with the value of biological control to the entire biosphere estimated at $417 billion per year [4]. More than 150 species of natural enemies are commercially available to be used in agricultural production [5]. The efficacy of new biocontrol agents needs to be assessed before they can be considered as part of an integrated pest management program and commercially produced.

Walnut (*Juglans regia* L.) cultivation is widespread around the world [6], but the walnut aphids, *Chromaphis juglandicola* (Kaltenbach), and dusky-veined aphid *Panaphis juglandis* (Goeze) have become major pests in walnut orchards where they reduce the quality and yield of nuts [7,8]. In recent years, these increasingly prevalent aphids have resulted in huge economic losses due to the reduction in tree vigor and nut size, yield, and quality [9,10]. The two aphid species colonize walnut trees at the same time with the relatively large aphid *P. juglandis* (adults 3.5–4.3 mm in length) feeding on the upper surface of leaves (adaxial) and the small aphid, *C. jugandicola* (1.2–2.3 mm) feeding on the lower surface of leaves (abaxial) [11]. Effective biological control of walnut aphids began in 1969 when the parasitoid wasp *Trioxys pallidus* Haliday was imported from Iran to California to control *C. juglandicola* [12]. But the competing primary and hyperparasitoids reduced the biocontrol efficacy of the parasitoids on aphids in the field. [13].

The anthocorid bug genus *Orius* Wolff has been regarded as a promising biological control agent for agricultural pests such as thrips, aphids, and mites [14,15,16]. *Orius sauteri* Poppius is a mass-produced biological control agent [17,18,19,20], which can be used to suppress many aphid species [18]. However, the efficacy of *O. sauteri* has not been assessed specifically for the control of aphids in walnut orchards. Here, we evaluate *O. sauteri* as a biological control agent against the two aphid species that are major pests of walnut trees: *C. juglandicola* and *P. juglandis*. This assessment will assist the integrated pest management program for walnut orchards.

## 2. Materials and Methods

### 2.1. Insect Collection

The laboratory experiments were carried out during 2019 and 2020; *P. juglandis* and *C. juglandicola* were collected from walnut orchards located in Yili (43°25′ N, 082°82′ E), China. The collected aphids were cultured on three-year-old walnut saplings for three generations under controlled conditions of 25 ± 2 °C temperature, 75–80% relative humidity, and 14 L:10 D (h) photoperiod. Adults of *O. sauteri* were purchased from a commercial supplier (Shandong Lubao Technology Development, www.saas-birc.com). The sexes of *O. sauteri* bugs used in experiments were not identified.

### 2.2. Laboratory Experiments

#### 2.2.1. The Functional Response of Predatory *O. sauteri* Bugs to Increasing Density of *P. juglandis* and *C. juglandicola* Aphids

The experiments were conducted in Petri dishes (9 cm in diameter) with a fresh walnut tree leaf cut to the size of the Petri dish, and moistened filter paper placed under the leaf. Individual *O. sauteri* adults were starved for 24 h, then supplied with 2nd–3rd instar aphids at six different prey densities (5, 10, 15, 20, 25, and 35). Six replicate experiments were conducted for each aphid species and density. After 24 h, the numbers of remaining prey were counted.

#### 2.2.2. Predator Preference for Prey Species

The Petri dish setup was as described above. To ensure that the two aphid species were of similar size, we used 1st instar nymphs of *P. juglandis* and 2nd–3rd instar nymphs of *C. juglandicola* in these experiments. Individual *O. sauteri* adults were starved as above, then supplied with 15 *P. juglandis* 1st instar nymphs and 15 *C. juglandicola* 2nd–3rd instar nymphs in one Petri dish. In this experiment, we cut the walnut leaves in half and to fit the Petri dish, a half leaf with the introduction of *P. juglandis* in the adaxial surface, and another half leaf with the introduction of *C. juglandicola* in the abaxial surface. This experiment was done in 4 in replications, with 6 Petri dishes in each replication.

#### 2.2.3. Biological Control Efficacy of Predatory *O. sauteri* against a Single Population of *P. juglandis* or *C. juglandicola*

Three levels of *O. sauteri* density (0, 1, and 2 adult bugs) were tested against prey aphids with each density replicated ten times. For each predator density, ten healthy walnut saplings each with six branches of the same size were selected. Five leaves at the top of each branch were used for the experiment. Ten 2nd–3rd instar *C. juglandicola* were placed on each of the five leaves on three branches and ten 2nd–3rd instar *P. juglandis* were placed on each of the five leaves of the remaining three branches per sapling. Either 0, 1, or 2 adults of *O. sauteri* were released on each branch and covered with a voile bag (50 cm long × 35 cm wide) (mesh size 0.3 mm). The number of aphids remaining from the initial 50 in each voile bag was counted every 3 days and ended after 15 days.

### 2.3. Field Experiments

#### 2.3.1. Biological Control Efficacy of Predatory *O. sauteri* against a Single Population of *P. juglandis* or *C. juglandicola*

Field experiments were carried out when walnut aphid populations reached peak abundance in walnut orchards located Yili County (43°25′ N, 082°82′ E) of the Xinjiang Autonomous Region, northwestern China. During the field experiments (15 July to 29 July 2020), the daily average temperature and humidity were 23 °C (13–33 °C) and 59%, respectively. No insecticide or other chemical was applied to the experimental fields. The biocontrol experiment on the single species and mixed populations were conducted simultaneously. Ten walnut trees of the same diameter trunk (10 cm) were randomly selected in a walnut orchard as experimental units. The experimental method was as described for the above laboratory experiments, i.e., five leaves at the end of six branches per tree had predatory bugs and prey aphids added as appropriate and covered by a voile bag.

#### 2.3.2. Biological Control Efficacy of Predatory *O. sauteri* against a Mixed Population of *P. juglandis* and *C. juglandicola*

The experimental unit was similar to that in the previous experiment except that only three branches of the same size were selected on each tree and 25 *P. juglandis* 2nd–3rd instars and 25 *C. juglandicola* 2nd–3rd instars were placed on the five leaves at the end of each branch, predators added and covered by a voile bag.

### 2.4. Data Analysis

The functional response of predatory *O. sauteri* bugs to increasing density of *P. juglandis* and *C. juglandicola*

The number of *P. juglandis* or *C. juglandicola* consumed after 24 h as a function of initial density was fitted to the Holling’s disc equation Type II [21,22].
(1)$$Na=\frac{a^{\prime }NT}{1+a^{\prime }ThN}$$

The characteristic of a Type II functional response is that predation increases at a decreasing rate with increasing prey density until a limit is reached [21]; *Na* represents the number of individual prey eaten by an individual predator per time unit, *N* is the initial number of prey offered to each predator at the beginning of the experiment, *a*’ is the rate of successful attack of prey by a predator, *T* is the total time of the experiment, and *Th* is the handling time.

A quadratic model was fitted to the functional response curve by non-linear least squares regression using Origin software to describe the effect of prey density on predation by an *O. sauteri* bug on the two aphid species. Predation data for each prey density were analyzed through the Kruskal-Wallis test and mean ranks were compared in pairwise comparison test at *p* < 0.05 because the data did not meet the assumptions of the normal distribution by the Kolmogorov-Smirnov test [23], even after transformation.

#### 2.4.1. Predator Preference for Prey Species

The preference of *O. sauteri* for one species of aphid prey over the other was estimated using the Cain Index [24]:(2)$$D=\frac{N_{P_{1}}\times N_{2}}{N_{P_{2}}\times N_{1}}$$
where *D* is a preference index for prey, *N_p_*_1_ and *N_p_*_2_ is the number of prey 1 (*P. juglandis*) or prey 2 (*C. juglandicola*) that were eaten, *N*_1_ and *N*_2_ are the initial numbers of prey 1 or prey 2. *D* > 1 indicates that the predator has a preference for prey 1.

Analysis of variance (LSD) was used to compare the differences among the mean number of aphids eaten by the *O. sauteri* adult in the four repeated experiments for each aphid species. When there was no significant difference, data from the four repeated experiments were combined for further analysis.

#### 2.4.2. Biological Control Efficacy of Predatory *O. sauteri* against *P. juglandis* and *C. juglandicola*

Three methods were used to evaluate the biological control efficacy of predatory *O. sauteri* bugs against *P. juglandis* and *C. juglandicola*. Firstly, control efficacy (percent control) was calculated by using the following equation [25]:(3)$$\begin{array}{cc} Control & efficacy \end{array}=(1-\frac{N_{1}}{N_{2}})\times 100\%$$

In Equation (3), *N*_1_ is the number of aphids in the treated group and *N*_2_ is the number of aphids in the untreated group at the end of the experiment.

Secondly, we used insect-days as an index of the efficacy of the predator. Cumulative insect-days were calculated by sequentially summing the individual insect-days. The insect-days formula for the area under this curve [26] was:(4)$$Insect-days=(X_{i+1}-X_{i})(\frac{Y_{i}+Y_{i+1}}{2})$$
where *X_i_* and *X_i+_*_1_ are adjacent points of time, and *Y_i_* and *Y_i_*_+1_ are the corresponding numbers of insects at those points in time. The percent reductions in cumulative insect-days indicate the percentage reduction between the treated and untreated groups.

Thirdly, we assumed that the growth of the aphid population was exponential within a limited window for the duration of the experiment. Population growth rate (PGR) was calculated using the following equation [27]:(5)$$PGR=\frac{\ln(N_{t})-\ln(N_{0})}{\Delta t}$$

*N_t_* is the number of aphids at the end of the experiment, *N*_0_ is the number of aphids at the beginning of the experiment, Δ*t* is the duration of the experiment (15 days), and ln is the natural logarithm.

In all statistical analyses, the data obtained from experimental groups were processed in SPSS 19. Most data of control efficacy, percent reduction in cumulative insect-days, and population growth rate (PGR) obtained from our experiments did not meet the assumptions of a normal distribution using the Kolmogorov-Smirnov test, even after transformation. Thus, the data from the experiments using different densities of predators (0, 1, or 2) were compared by analysis of variance using the Kruskal-Wallis test with a *p* = 0.05 level of significance.

## 3. Results

### 3.1. The Functional Response of Predatory O. sauteri Bugs to Increasing Density of P. juglandis and C. juglandicola

The number of prey consumed by the predatory bug increased with an increase in prey density. Overall, *O. sauteri* adults consumed more 2nd–3rd instars of *C. juglandicola* than of *P. juglandis* at each density. There was a significant difference in the number of *C. juglandicola* consumed when prey density was 5 or 10 per leaf compared to when prey density was 35 per leaf. For *P. juglandis*, there was a significant difference in the number of prey consumed at prey densities of 5 or 10 per leaf compared to 25 prey per leaf (Table 1).

The functional response of predatory *O. sauteri* towards *C. juglandicola* and *P. juglandis* fitted the Type II response modeled by the Holling’s disc equation from the quadratic models fitted to the functional response curves by non-linear least squares regression (*R*^2^ = 0.82 and *R*^2^ = 0.78, respectively). When the prey was *C. juglandicola*, the handling time (*Th*) was 1.73 h, the rate of successful attack (*a*’) was 1.06, and the theoretical maximum predation was 14 aphid nymphs over 24 h. When the prey was the larger *P. juglandis*, the *Th* was 2.01 h, *a*’ was 1.01, and the theoretical maximum predation rate was 12 aphid nymphs over 24 h.

### 3.2. Predator Preference for Prey Species

The number of *P. juglandis* and *C. juglandicola* eaten by the adult *O. sauteri* was not significantly different among the four repetitions of the experiment (*F* = 1.75, *df* = 3, *p* > 0.05 for *P. juglandis*; *F* = 2.17, *df* = 3, *p* > 0.05 for *C. juglandicola*), so data were pooled. More *P. juglandis* nymphs than *C. juglandicola* nymphs were eaten by *O. sauteri*. The Cain Index (*D*) was more than 1 indicating that *O. sauteri* preferred *P. juglandis* (Table 2).

### 3.3. Biological Control Efficacy of Predatory O. sauteri against a Single Population of P. juglandis or C. juglandicola in the Laboratory and Field

In both laboratory and field experiments, *O. sauteri* effectively controlled populations of each aphid species. In the untreated groups in the laboratory and field, the populations of *P. juglandis* began to increase rapidly on the ninth day, but after 15 days the population size in the laboratory was less than in the field populations (27 ± 50 versus 486 ± 65, respectively). The laboratory populations of *C. juglandicola* began to increase rapidly on the sixth day, but the field populations began to reproduce on the third day; after 15 days, the mean number of aphids in the laboratory populations was greater than in the field populations (1610 ± 227 versus 534 ± 39, respectively) (Figure 1).

The populations of *P. juglandis*, as well as *C. juglandicola,* declined after 15 days if treated with predators. The mean control efficacy of *P. juglandis* were 79% ± 8.7% with 1 predator and 92% ± 4.3% with 2 predators in the laboratory while in the field the means were 54% ± 8.7% and 84% ± 6.1% with 1 or 2 predators. In the case of *C. juglandicola*, the mean control efficacy was 63% ± 13.2% and 95% ± 4.6% with 1 or 2 predators in the laboratory but in the field, the means were 76% ± 6.1% and 90% ± 2.3% with 1 or 2 predators. The mean control efficacy of *O. sauteri* was highest by using these two predator densities (*p* < 0.05). The percent reduction of cumulative insect-days of populations with predatory bugs compared to control populations with no predatory bug(s) also indicated that *O. sauteri* suppressed the population size of *P. juglandis* and *C. juglandicola*. The mean percent reduction of *P. juglandis* and *C. juglandicola* were greater in the laboratory experiments than in the field experiments (*P. juglandis**: p* < 0.05; *C. juglandicola**: p* < 0.05). In the laboratory, *P. juglandis* exhibited a mean percent reduction of cumulative insect-days which were 67% ± 6.2% and 78% ± 4.9% with 1 or 2 predators. The mean percent reduction of cumulative insect-days of *C. juglandicola* were 70% ± 9.7% and 93% ± 3.3% with 1 or 2 predators. Field experiments revealed that the mean of *P. juglandis* were 44% ± 5.5% and 66% ± 4.6% with 1 or 2 predators but for *C. juglandicola* the mean was 54% ± 7.3% and 71% ± 4.4% with 1 or 2 predators. Two adult *O. sauteri* were more effective than one adult *O. sauteri*. (*p* < 0.05) (Figure 2).

The mean population growth rates (PGR) of *P. juglandis* were 0.1 ± 0.01, −0.06 ± 0.03, and −0.12 ± 0.03 with 0, 1, or 2 predators in the laboratory, while in the field they were 0.15 ± 0.01, 0.08 ± 0.02, and −0.03 ± 0.03 with 0, 1, or 2 predators. In the case of *C. juglandicola*, the mean growth rates were 0.22 ± 0.01, 0.07 ± 0.05, and −0.11 ± 0.03 with 0, 1, or 2 predators in the laboratory, and in the field, the mean were 0.16 ± 0.01, 0.05 ± 0.01, and −0.02 ± 0.02 with 0, 1, or 2 predators. The mean population growth rates (PGR) were significantly different in the presence and absence of different initial densities of *O. sauteri* bugs (*H* = 19.1, *df* = 2, *p* < 0.05 for *P. juglandis* in the laboratory; *H* = 18.1, *df* = 2, *p* < 0.05 for *C. juglandicola* in the laboratory; *H* = 18.2, *df* = 2, *p* < 0.05 for *P. juglandis* in the field; *H* = 21.4, *df* = 2, *p* < 0.05 for *P. juglandis* in the field). Aphid populations treated with two adult *O. sauteri* had lower PGRs than those treated with one. The PGR of *P. juglandis* populations (−0.12 ± 0.03) was less than *C. juglandicola* (−0.11 ± 0.03) in laboratory experiments. Conversely, the PGR of the *P. juglandis* population (−0.03 ± 0.02) was greater than *C. juglandicola* (−0.02 ± 0.02) treated with two predatory bugs in field experiments (Figure 2).

### 3.4. Biological Control Efficacy of Predatory O. sauteri against a Mixed Population of P. juglandis and C. juglandicola Aphids in the Open Field

Overall, exposure of a mixed population of *P. juglandis* and *C. juglandicola* aphids to one or two *O. sauteri* predatory bugs resulted in increased control efficacy, increased percent reduction in cumulative insect-days, and decreased PGRs compared to control populations without bug(s) in a trend similar to experiments using single-species populations in the field (Figure 2). Releasing two *O. sauteri* resulted in better control than using one. However, the bio-control efficacy of one and two predatory bug(s) against *P. juglandis* and *C. juglandicola* aphids in a mixed population was different for the two species over time (Figure 1). After 15 days exposed to two predatory bugs, the mean population of *P. juglandis* was 107 ± 22, higher than the mean number of 41 ± 8 per *C. juglandicola* population. The control efficacy and the percent reduction in cumulative insect-days resulting from exposure of a mixed population of the two aphid species to one or two predatory *O. sauteri* bugs were lower than for the same species of aphid in a single population (Figure 2). In addition, the mean PGRs of the two aphid species in mixed populations were higher than for the two species in single populations when exposed to one or two predatory bugs. In the treatments with two *O. sauteri* against a mixed aphid population, the control efficacy of *C. juglandicola* was significantly higher (71% ± 11.5%) than when one *O. sauteri* was released (39% ± 18.3%). The control efficiency of *P. juglandis* was similar for one (42% ± 12.8%) and two (39% ± 18.3%) predator treatments as well.

## 4. Discussion

Our experiments revealed that *O. sauteri* adults were effective in suppressing populations of the two walnut aphids *P. juglandis* and *C. juglandicola*. Functional responses showed the predator had a strong capacity for consuming aphids of the two species in both laboratory and open field experiments.

The control efficacy of *O. sauteri* against *P. juglandis* and *C. juglandicola* aphids was greater than has been found against other aphid pests. Under laboratory conditions, the theoretical predation capacity of 4–5th instar *O. sauteri* on *Aphis glycine* (Matsumura) was up to 8 per day [28], and on *Brevicoryne brassicae* (Linnaeus) adults it was 6 per day [29]. We found that *O. sauteri* consumed approximately 12 *P. juglandis* or 14 *C. juglandicola* 2nd–3rd instar aphids within 24 h. Furthermore, the two aphid species were quite different in size (e.g., the 2nd–3rd instar of *P. juglandis* was larger than *C. juglandicola*) [30]. So, although slightly more *C. juglandicola* were eaten, the biomass of *P. juglandis* consumed was greater.

Comparing the biological control efficacy of predatory *O. sauteri* against populations of a single species of walnut aphid, the decline in aphid numbers was greater in laboratory experiments than in open field experiments suggesting environmental factors can influence efficacy; abiotic variables can affect predation of predatory bug populations, such as the change in temperature, humidity, wind, and rain. [31,32]. The laboratory conditions with stable climate led to consumption increasing by predators [33]. By contrast, during our field study, the highest temperature was 33 °C at noon and the lowest was 14 °C at night, predatory capacity can be negatively affected by extreme temperature [34], and the temperature fluctuation in the field also influenced the population dynamics of predators [35]. In addition, wind interfering with the movement and sensing of olfactory cues might affect predator feeding performance and ultimately cause a decline in the biocontrol efficacy [36].

Biotic factors, including prey insects, can determine the biocontrol efficacy of predators. *P**. juglandis* was consumed more often by *O. sauteri* than *C. juglandicola* when both were provided in laboratory prey choice experiments. However, the effect of this preference for *P. juglandis* over *C. juglandicola* was not always reflected in our field experiments. For example, when mixed populations of aphids (equal numbers of both species) were exposed to *O. sauteri,* the number of *P. juglandis* was greater than the numbers of *C. juglandicola* in the populations after 15 days. The preferred orientation of aphids on their host tree due to insect phototaxis may explain this result. Both the predatory bug *O. sauteri* and *C. juglandicola* aphids prefer to inhabit the lower surface of walnut tree leaves thus increasing the likelihood of encounters between the two. *Orius sauteri* are sensitive to light and distribute on the lower surface of leaves [37]. Conversely, *P. juglandis* aphids are found on the adaxial surface of leaves and are less likely to come into contact with *O. sauteri*.

The density of aphid prey affected the biocontrol of predators in our study. The number of aphids consumed by the predatory bug in one day increased with increasing prey density, as has been found for other predatory bugs: *Xylocoris flavipes* (Reuter) against the beetles *Tribolium castaneum* (Herbst) and *Attagenus megatoma* (F.) [38], as well as *Macrolophus pygmaeus* (Rambur) against *Myzus persicae* (Sulzer) [39,40]. The biocontrol efficacy of *O. sauteri* against each aphid species in a mixed-species population was less than in a single-species population. Population size dropped significantly when two species were mixed on the same leaf compared with single populations of each aphid species. Studies have shown competition for food and space between *C. juglandicola* and *P. juglandis* [41], in mixed-species aphid populations, there may be competition among individuals of the two aphid species leading to a decline of population size compared to single-species populations. Biocontrol efficacy of *O. sauteri* declined in the mixed group due to the lower density, with lower prey capacity determined by the competition among two coexisting aphids in our study.

All three approaches to measuring changes in aphid prey population used in our study described the biological control efficacy of *O. sauteri* bugs well. Control efficacy, in a classic approach, focuses on the final number of aphids exposed to the predator relative to the predator-free group regardless of the timing. However, the cumulative insect-days served as an index of the overall effectiveness of the predator; the integrated area under the curve of insect numbers over time described the biological control function or effectiveness of the predator. Both the abundance of pest aphids remaining, and infestation duration (timing) were considered [28]. In addition, the PGR of aphids can be used as an indicator of the efficacy of a biological control agent. The initial and final abundance of aphids and the time interval were used for the calculation of PGR (see Equation (5)). The lower the PGR the higher the bio-control efficacy (negative values of PGR are best) and the greater the reduction of insect-days. The combination of these three approaches should be considered when assessing bio-control practices.

This is the first report of the biocontrol of *O. sauteri* on two walnut aphid species based on both laboratory and field experiments. Further studies are needed to evaluate the control efficacy of *O. sauteri* in the context of a practical economic threshold for aphid infestation in orchard situations.

## 5. Conclusions

*Orius sauteri* can be a potential biological control agent for *P. juglandis* and *C. juglandicola* in walnut orchards. The functional response reflects the high consumption of the predator against two aphid species. The predator’s prey preference indicated *P. juglandis* was preferred, nevertheless, it was more effective in controlling *C. juglandicola* in the mixed aphid populations in the field. Control efficacy against single populations of the two aphid species was higher in laboratory experiments than in the open field. In the open field experiments, efficacy declined in mixed populations compared to single populations, particularly for the larger walnut aphid (*P. juglandis*) due to behavioral differences and competition between the two aphid species. Successful control may require higher release rates when both species are present.

## Figures and Tables

**Figure 1 insects-12-00025-f001:**
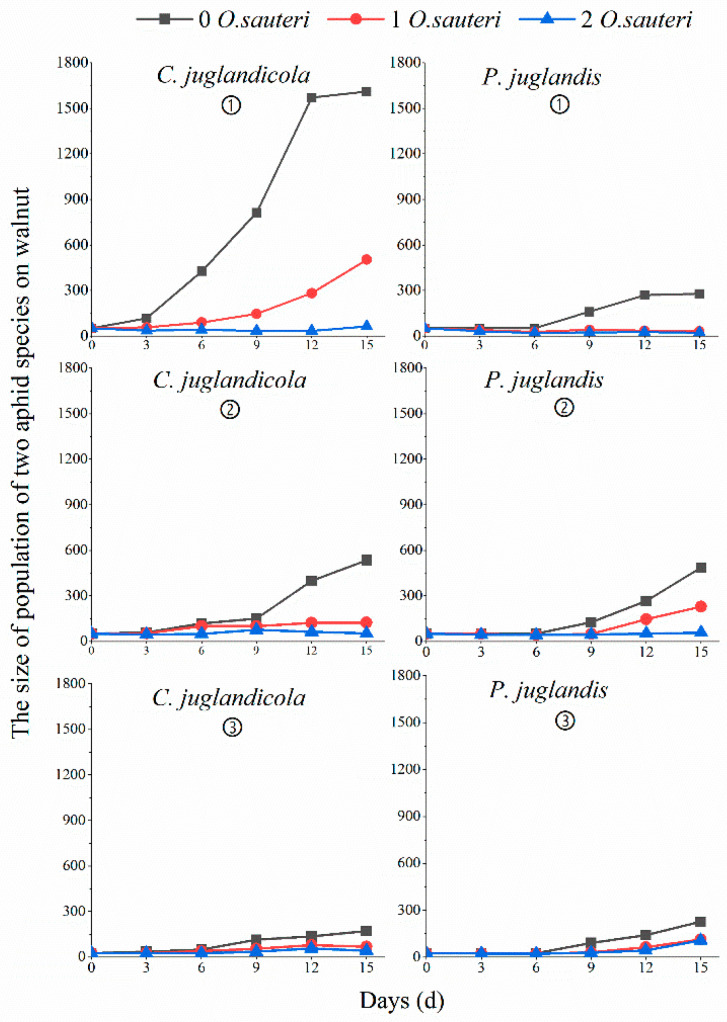
The mean number of aphids in populations of two aphid species exposed to an initial density of 0, 1, and 2 *O. sauteri* predatory bugs over 15 days. Note: ① indicates the laboratory experiments, ② indicates the field experiments, ③ indicates the mix population experiments.

**Figure 2 insects-12-00025-f002:**
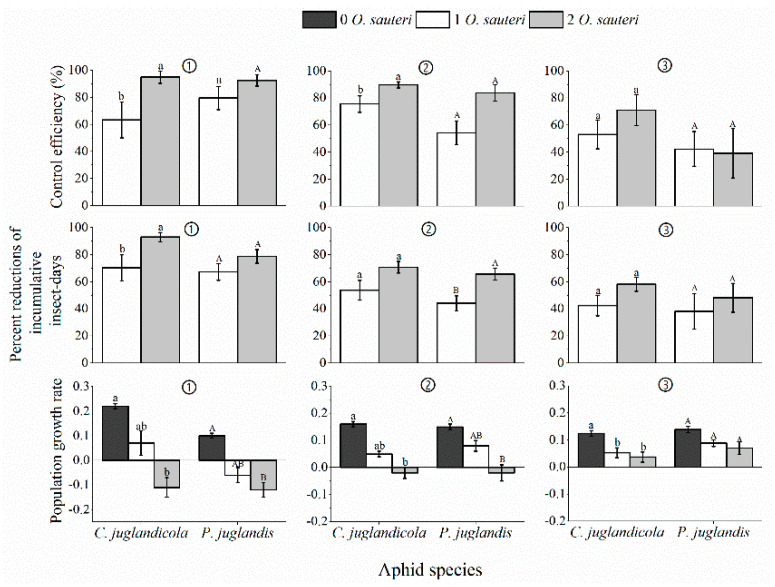
The control efficacy, percent reduction of cumulative insect-days, and population growth rate of two aphid species exposed to an initial density of 0, 1, and 2 *O. sauteri* predatory bugs over 15 days. Note: Different lowercase letters a and b indicate significant differences at the level of *p* < 0.05 for *C. juglandicola*; different capital letters A and B indicate significant differences at the level of *p* < 0.05 for *P. juglandis*; ① indicates the laboratory experiments, ② indicates the field experiments, ③ indicates mix population experiments.

**Table 1 insects-12-00025-t001:** Mean number of 2nd–3rd instar of *Panaphis juglandis* and *Chromaphis juglandicola* aphids preyed on by an *Orius sauteri* adult bug on walnut leaf disks over 24 h at 25 ± 2 °C.

Prey	Prey Density (Aphids/Leaf Disc)					
	5	10	15	20	25	35
*Chromaphis juglandicola*	4.2 ± 0.7 ^bc^	4.8 ± 1.3 ^b^	6.2 ± 1.1 ^ab^	9.0 ± 1.1 ^ab^	11.5 ± 0.7 ^ab^	11.7 ± 2.2 ^a^
*Panaphis juglandis*	3.8 ± 0.3 ^c^	4.0 ± 0.3 ^c^	6.5 ± 1.3 ^abc^	8.5 ± 1.3 ^abc^	8.8 ± 0.5 ^ab^	9.2 ± 1.6 ^abc^

Note: Different lowercase letters indicate significant differences at the level of *p* < 0.05.

**Table 2 insects-12-00025-t002:** Prey choice of predatory *O. sauteri* bug against two walnut aphid species.

*P. juglandis*		*C. juglandicola*		Cain Index
Density	Predation (*N_p_*_1_)	Density	Predation (*N_p_*_2_)	
15	6.87 ± 0.47	15	4.25 ± 0.51	1.82 ± 0.22

## Data Availability

There is no additional data to disclose, all data are included in this manuscript.

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
