# Peer review of "Is Orius sauteri Poppius a Promising Biological Control Agent for Walnut Aphids? An Assessment from the Laboratory to Field"

_insects, 2021, doi:10.3390/insects12010025_

Round 1

Reviewer 1 Report

Overall, I think the introduction needs a bit more detail. For example, the authors mention a successful parasitoid that suppresses one of the aphids, but they do not mention if that is part of their bio control program, or why it is not. Does it not work in China? That is very possible, but there is not indication that it has been tried. The introduction goes from talking about walnut aphids being major pests to mentioning that they can be controlled by a parasitoid. The intro is not fleshed out enough and is disjointed. There are also many instances of papers appearing to be cited where they shouldn't and sentences that need a lot more explanation and citations. 

The laboratory petri dish experiments are lacking controls and need to be redone with these added to make any comparisons valid.

The data analyses are not well explained and need some citations and rationales. The different experiments are described seem like several of them could be combined and may not be tested what they are described as testing.

The experiments should have been combined probably from the start to really get controls that were comparable. I am also not sure that the data analysis for this experiment was appropriate.

I had many comments on the manuscript. There were no line numbers so I added comments throughout the PDF. 

Author Response

Response to Reviewer 1 Comments

We are grateful for the insightful comments on and valuable improvements on our paper. We have incorporated most of your suggestions. All changes are highlighted within the previous manuscript. Please see below with blue font for a point-by-point response to your comments and concerns.

Additionally, we added the line numbers in revised manuscript for easy reading and reviewing. We processed your comments in two ways. All of your concerns were processed in the below point to point. The other way was done in word file in attachment.  

Your concered questions

Point 1: (In line 56) Add the common names here. Since they are not used again thoughout the manuscript, you can put them in parentheses and not mention them again. As a result, take them out of the other sentences in the introduction.

Response 1: Common names were edited here, and common names were ignored in the later section of this manuscript.

Point 2: (In line 59) Please add an estimate of losses due to aphids here. Also, one of your citations here is from 1982, which is not recent, so please change the wording here and add some more detail about the economic losses due to these aphids.

Response 2: We have changed the wording and citation here and explained in more detail. As follows: 'In recent years these increasingly prevalent aphids have resulted in huge economic losses by reduces tree vigour and nut size, yield and quality [7-9]'.

Point 3: (In line 59) This sentence could be more clearly worded. I would suggest the following, with the parentheses being the sizes: The two aphid species colonize walnut trees and the same time with the relatively large aphid, P. juglandis (...) feeding on the upper surface of leaves and the small aphid, C. jugandicola (....) feeding on the lower surface of leaves.

Response 3: We have modified this sentence as the following: 'The two aphid species colonize walnut trees and the same time with the relatively large aphid, P. juglandis (adults 3.5-4.3 mm in length) feeding on the upper surface of leaves (adaxial) and the small aphid, C. jugandicola (1.2-2.3mm) feeding on the lower surface of leaves (abaxial)'.

Point 4: (In line 64) If long term control has been achieved since 1969, why do we need to determine if this predator is effective for it? I believe that it is necessary, I would just like some further justification from the authors. It is mentioned in the next paragraph that C. juglandicola is a major pest, but if it is being controlled by a parasitoid, then why isnt that being used here.

Response 4: Changed as  'But competing primary and hyperparasitoids were concerned on the biocontrol efficacy of parasitoid on aphids in the field. [12]'.

Point 5: (In line 68) This paper mentions that O. sauteri is available as a bio control agent but that is in the introduction without a citation and that is not what this paper is about. Please find a citation that is about O. sauteri being a mass produced biological control agent.

Response 5: We changed citations [16-19] that is about O. sauteri being a mass produced biological control agent.

Point 6: (In line 71) Remove the common names here.

Response 6: Removed.

Point 7: (In line 75)Change 2010 to 2020

Response 7: Revised as to 2020.

Point 8: (In line 83) Make sure you change your headings to italics or something like it to distinguish them from the text.

Response 8: The headline style was revised in line with your suggestion.

Point 9: (In line 88) Move this sentence to the beginning of the paragraph. Also, be careful with font sizes and make sure you are being consistent.

Response 9: The font size has been adjusted to the same size.

Point 10: (In line 89) In the petri dish experiments, did you have any petri dishes with no predators present to control for possible aphid escape? If so, please add those here. If not, why not?

Response 10: We did not have control group. Really, there has not control treatment in experiment of the function response usaully. We supported some citations to confirm this above point.

  1. Lacava, M.; García, L.F.; Viera, C.; Michalko, R. The pest-specific effects of glyphosate on functional response of a wolf spider. Chemosphere 2020:127785. Doi:org/10.1016/j.chemosphere.2020.127785.
  2. Faria, L.D.; Trinca, L.A.; Godoy, W.A. Cannibalistic Behavior and Functional Response in Chrysomya albiceps (Diptera: Calliphoridae). Journal of Insect Behavior 2004, 17, 251-261.Doi:10.1023/B:JOIR.0000028574.91062.18

Point 11: (In line 97) What does each experiment mean? It seems like you had 30 aphids in petri dishes with starved Orius and you 24 dishes of these. It seems like one experiment with no treatments. Did you have a control with no predators? Did you compare it to one with 30 of a single species? Or is this all part of the above experiment? Either way, there do not seem to be any controls to compare things to.

Response 11: The purpose of this experiment was to clarify the prey choice of predator among two aphid species through this simple experiment, no control group are not needed in prey choice experiment. We have attached some citations to support our study.

   1. Schuler, T.H.; Clark, A.J.; Clark, S.J.; et al. Laboratory studies of the effects of reduced prey choice caused by Bt plants on a predatory insect. Bulletin of Entomological Research 2005, 95, 243-247.Dio: 10.1079/BER2004356.

   2. Haddaway, N.R.; Wilcox, R.H.; Heptonstall, R.E.A.; Griffiths, H.M.;  Mortimer, R.J.G; Christmas, M.; Dunn, A.M. Predatory functional response and prey choice identify predation differences between Native/Invasive and Parasitised/ Unparasitised Crayfish. Plos One 2012, 7, 1-8. Dio: 10.1371/journal.pone.0032229.

Point 12: (In line 109) Make sure your headings are in a different font style.

Response 12: We have changed the headline style.

Point 13: (In line 139) This citation does not appear in the list and should be amended.

Response 13: Changed as the following: ‘A quadratic model was fitted to the functional response curve by non-linear least squares regression by Origin to describe the effect of prey density on predation by an O. sauteri bug on the two aphid species.’

Point 14: (In line 143) Add a citation here for this test.

Response 14: Added a citation [20] here.

Point 15: (In line 144) This section is confusing to me. What does it mean that it was not normal here? Is this not the correct way to analyze these data then? Also, did you fit the other models and they did not fit or was it just assumed it was a Type II. You talk a lot of about predation rate, but you never have predation rate in a table. you have average number eaten in a table, but you should have rate there too so the reader does not have to do math. By the numbers in your table, the rate decreases a lot with more prey items available as it is very high when only a few are available.

Response 15: In date analyze, independent variables (predator consumption under different prey densities) do not fit normal distribution, so we use KW test here. We offer the detailed information on this question. Please see the updated context. Generally, Type II can be used to describe the predator potential capacity for consuming the biomass of prey theoretically. We use this model to describe the attacking capacity of this predator bug on two species aphids, respectively. The result from this function experiment, combined with lab and field experiments, presented the bio-control efficacy of this predatory bug in different scale or level.

Point 16: (In line 151) Indicate what prey 1 and prey 2 are. Be explicit.

Response 16: More details were offered as the following: 'Where D is a preference index for prey, Np1 and Np2 is the number of prey 1 (P. juglandis) or prey 2 (C. juglandicola) eaten, N1 and N2 are the initial number of prey 1 or prey 2. >1 indicates that the predator has a preference for prey 1.'

Point 17: (In line 156) What were the treatments in these experiments? Were these the petri dishes that had 15 of each species?

Response 17: Yes, the headline in this section is: 2.6. Predator preference for prey species.

Point 18: (In line 160) If this is standard, please add a citation for where it comes from.

Response 18: Modified and added citation [22].

Point 19: (In line 161)What is the treatment group? Does this mean the group that had any predators added? Is this both of the species of aphids?, Is N2 then just the branches with no Orius?

Response 19: We have changed the wording here and explained in more detail. In equation 1, N1 is the number of aphids in the treatment group with O. sauteri density, N2 is the number of aphids at untreated group without O. sauteri. N1 and N2 showed the aphid population size at the end of the experiment in both aphid species.

Point 20: (In line 167) I think there needs to be a lot more detail for this equation. Also, a citation for this. Where does it come from?

Response 20: The conventional equation is common and from this paper in reference 23..

Point 21: (In line 174) Need a citation for this equation too or more explanation for the use of natural log.

Response 21: We have modified and added citation [24] .LN(Nt) can be showed as log2.718(Nt)

Point 22: (In line 184)If everything was not normal and there were negative numbers in the growth rate, perhaps the equations and analyses being used were not correct in the first place.

Response 22: Changed as the following: “In date analyze, independent variables do not fit normal distribution, so we use KW test here. Updated information in the first place as the following; 'A quadratic model was fitted to the functional response curve by non-linear least squares regression by Origin to describe the effect of prey density on predation by an O. sauteri bug on the two aphid species. The data of predator consumption under different prey densities did not meet the assumptions of the normal distribution by the Kolmogorov-Smirnov test [20], even after transformation. A nonparametric Kruskal-Wallis test (P< 0.05) was used to compare differences between the mean ranks of the predation data for each prey density. “

Point 23: (In line 189)You are testing these as a type II response. You need a figure that shows that response here. Some indication that it fits. Their fittness can be seen in the next paragraph.

Response 23: The main target of our study was to understand the bio-control efficacy of predator. Lab-field experiments, combinated with the function repose experiment(result from Type II) were conducted.. The figure is not necessary here because the function repose experiment is not key experiment in our study.

Point 24: (In line 192) Sure, but the rate is what you were testing and is that different? Of course the average numbers are since there are so many more aphids available to the predators with no competition with other predators.

Response 24: Not necessary in here. The function response experiment was used to support the lab and field biocontrol experiment. Same question in line 189.

Point 25: (In line 200) Prove this with a figure and/or the rates in a table

Response 25: Not necessary to present the result with figure.

Point 26: (In line 201) Rate of increase is not the same as higher predation rate. the predation rate was lower

Response 26: I deleted these sentences for consistence in all of context.

Point 27: (In line 214) You can remove prey 1 and prey 2 as it will have already been specified earlier in the manuscript.

Response 27: We have removed prey 1 and prey 2, and they were mentioned in the previous data analysis.

Point 28: (In line 217) Add 'of' here

Response 28: Modified.

Point 29: (In line 242) Why is this a broken axis when it is the same as the one above it? These should all be the same as they are really hard to interpret as they are presented. Remove the broken axis.

Response 29: We have changed  figures and uploaded it.

Point 30: (In line 253) Instead of the numbers here, label them the same as the line graphs. Or, use the numbers on the line graphs. Either way, just be consistent.

Response 30: We used the numbers on the line graphs.

Point 31: (In line 256) The H test only tells you that there is a difference between the groups not where the differences are.

Point 32: (In line 264) What were the mean ranks for these?

Point 33: (In line 267) You need statistics here. Is this a significant difference?

Point 34: (In line 267) Same with these. Are they significantly different.

Point 35: (In line 267) Since the H test cannot establish which groups have significant differences, another post how may be appropriate to use to detemine which groups have the significant differences.

Response 31-35: Questions from line 256-267 were processed as the following:As follows: ‘The mean population growth rates(PGR) of P. juglandis were 0.1±0.01, -0.06±0.03 and -0.12±0.03 with 0, 1 or 2 predators in laboratory while in the field the mean were 0.15±0.01, 0.08±0.02 and -0.03±0.03 with 0, 1 or 2 predators. In case of C. juglandicola, mean were 0.22±0.01, 0.07±0.05 and -0.11±0.03 with 0, 1 or 2 predators in laboratory and in the field the mean were 0.16±0.01, 0.05±0.01 and -0.02±0.02 with 0, 1 or 2 predators. The mean population growth rates(PGR) were significantly different in the presence and absence of different initial densities O. sauteri bugs (H=19.1, df=2, P<0.05 for P. juglandis in laboratory; H=18.1, df=2, P<0.05 for C. juglandicola in laboratory; H=18.2, df=2, P<0.05 for P. juglandis in field; H=21.4, df=2, P<0.05 for P. juglandis in field). Aphid populations treated with two adult O. sauteri had lower PGRs than those treated with one. The PGR of P. juglandis populations (-0.12±0.03) was less than C. juglandicola (-0.11±0.03) in laboratory experiments. Conversely, the PGR of P. juglandis populations (-0.03±0.02) was greater than C. juglandicola (-0.02±0.02) treated with two predatory bugs in field experiments (Fig. 2).’

Point 36: (In line 301) A little more detail here. Why do you think that Orius would have been better at aphid predation in the lab vs outside since in both cases they were in bags on branches? Is this just the effect of being in the lab, which can be very hard on experimental animals and may have affected the aphids more. I would have expected a greater aphid decline outside vs the lab given aphids are often pretty sensitive to heat. I think this point needs more exploring because it is really the opposite of what you would expect.

Point 37: (In line 307) as has been shown in previous publications.

Point 38: (In line 308) These are all good points, but do not really have much to do with why you saw more decline in the lab vs the field. Unless you kept your lab at 35-37 C, this information would still suggest the field should have resulted in lower aphid numbers. Also, since all your experiments were in bags, the temperature in the bags outside may have been hotter so again, that would have resulted in the opposite of what you report. I believe your results, you just need to try to explain them.

Response 36-38: Bigger changes as the following:Comparing the biological control efficacy of predatory O. sauteri against populations of a single species of walnut aphid, the decline in aphid numbers was greater in laboratory experiments than in open field experiments suggesting environment factors can influence efficacy; abiotic variables can affect predation of predatory bug populations, such as change in temperature, humidity, wind and rain. [28,29]. The laboratory conditions with stable climate lead to more intense predation pressure [30]. By contrast, during our field study, the highest temperature was 33℃at noon and the lowest was 14℃at night, predator populations can be negatively affected by extreme temperatures that affect their consumption ability [31], and temperature change in the field impact predators more than lower trophic levels and alter the population dynamics and the sustainability of the ecosystem [32]. In addition, wind interfering the movement and sense of olfactory cue might be affect the predator feeding performance and decline the bio-control efficacy [33].

Reviewer 2 Report

The manuscript is in general rather well written. Some minor changes / corrections are nevertheless needed before acceptance.

when reportig statistic of analysis of variance, being F a ratio, 2 df must be reported (section 3.2)

Table 1: report in caption the statistic analysis used and check data (is 8.8 correct for prey density = 25? If so the letter "a" is likely incorrect)

Figure 1: normalise Y axis using the same value for each graph

Check the use of italic for scientific names

The last sentece in discussion section is likely redundant.

Author Response

Response to Reviewer 2 Comments

We are grateful for the insightful comments on and valuable improvements on our paper. We have incorporated most of your suggestions. All changes are highlighted within the previous manuscript. Please see below with blue font for a point-by-point response to your comments and concerns.

Additionally, we added the line numbers in revised manuscript for easy reading and reviewing. We processed your comments in two ways. All of your concerns were processed in the below point to point. The other way was done word file in attachment.  

Point 1: (In line 46) Page 1:  ……ecosystem service substantially contributing to protecting….

Response 1: Changed as ‘The biological control of pests by their natural enemies is a crucial ecosystem service  substantially contributes to protecting the environment as well as maintaining biodiversity [1]. The efficacy of natural enemies are available commercially for protection of crops and trees [2]. ’ From line 46 to row 48.

Point 2: (In line 72) Page 2:  Last sentence of introduction is awkward – will inform the development – please re-word.

Response 2: Revised as ‘This assessment will inform the package of an integrated pest management program for walnut orchards.’ From line 71 to row 72.

Point 3: (In line 80) The sex of O. sauteri bugs used in experiments was not determined.

Response 3: We mention that in first paragraph in material and method: ‘The sex of O. sauteri bugs used in experiments was not identified.’ 

Point 4: (In line 94) Lab experiments:  I was curious about the leaf surface orientation for this experiment:  which leaf surface (abaxial or adaxial) was oriented upward in the petri dish?  After learning, in the Discussion, that aphid preference for leaf surface is important, more detail must be provided here.  You potentially had one of the two aphid species feeding on a non-preferred leaf surface and a predator searching on a non-preferred leaf surface oriented opposite of its field preference.  Please provide more detail.

Response 4: Sorry for poor description in the method before. We offered more explanation in here. have modified and explained in more detail, as follow: In this experiment we cut the walnut leaves equally in half which were the same size with petri dish, a half leaf with inocation P. juglandis in the adaxial surface, and another half leaf with inocation C. juglandicola in abaxial surface.

Point 5: Page 5:  Table 1.  Please look at prey density for P. juglandis at 25 and 35. The mean separation letters provide for these two numbers do not make sense.  How is 9.2 an ‘ab’ but 8.8 is an ‘a’??

Response 5: We have rewritten the mean separation letters in Table 1.

Point 6: (In line 198) Paragraph below Table 1.  You did not test/analyze the ‘rate of increase’ so comments about more/less should not be made.

Response 6: Deleted these sentences.

Point 7: (In line 72) Paragraph above Table 2.  You switch from saying eaten to nymphs were ‘killed’.  Was there a reason?  Being eaten vs. being killed are different and would have different implications regarding these results.

Response 7: We have changed ‘killed’ into eaten. From line 208.

Point 8: (In line 217) Last paragraph of the page:  …….effectively controlled populations of each aphid…….

Response 8: We have changed the wording as your comments in line 217.

Point 9: (In line 167) Throughout the manuscript: Most, if not all, instances of ‘percent’ should be ‘percentage’.

Response 9: This is a fixed terminology in ‘ percent reduction of cumulative insect-days ’. See the reference in 23.

Point 10: (In line 225) Page 6, first full paragraph. Any statistics to support comment such as ‘control efficacy was greatest’?

Response 10: We have rewritten this paragraph, as follows: ‘The populations of P. juglandis, as well as C. juglandicola, declined after 15 days if treated with predators. The mean control efficacy of P. juglandis were 79%±8.7% with 1 predator and 92%±4.3% with 2 predators in laboratory while in the field the means were 54%±8.7% and 84%±6.1% with 1 or 2 predators). In case of C. juglandicola the mean control efficacy were 63%±13.2% and 95%±4.6% with 1 or 2 predators in laboratory but in the field the mean were 76%±6.1% and 90%±2.3% with 1 or 2 predators. The mean control efficacy of O. sauteri were highest by using these two predator densities.’

Point 11: (In line 225) Page 7, first full paragraph.  Delete ‘populations’ following C. juglandicola. Were the populations growth rates statistically significant?

Response 11: We have deleted ‘populations’ following C. juglandicola in full paragraph, from line 217 to 223. And, the populations growth rates were statistically significant in the presence and absence of different initial densities O. sauteri bugs. We rewrite this paragraph as follows ‘The mean population growth rates(PGR) of P. juglandis were 0.1±0.01, -0.06±0.03 and -0.12±0.03 with 0, 1 or 2 predators in laboratory while in the field the mean were 0.15±0.01, 0.08±0.02 and -0.03±0.03 with 0, 1 or 2 predators. In case of C. juglandicola, mean were 0.22±0.01, 0.07±0.05 and -0.11±0.03 with 0, 1 or 2 predators in laboratory and in the field the mean were 0.16±0.01, 0.05±0.01 and -0.02±0.02 with 0, 1 or 2 predators. The mean population growth rates(PGR) were significantly different in the presence and absence of different initial densities O. sauteri bugs (H=19.1, df=2, P<0.05 for P. juglandis in laboratory; H=18.1, df=2, P<0.05 for C. juglandicola in laboratory; H=18.2, df=2, P<0.05 for P. juglandis in field; H=21.4, df=2, P<0.05 for P. juglandis in field). Aphid populations treated with two adult O. sauteri had lower PGRs than those treated with one. The PGR of P. juglandis populations (-0.12±0.03) was less than C. juglandicola (-0.11±0.03) in laboratory experiments. Conversely, the PGR of P. juglandis populations (-0.03±0.02) was greater than C. juglandicola (-0.02±0.02) treated with two predatory bugs in field experiments (Fig. 2).’ 

Point 12: Section 3.4 seems to be a repeat of what has been stated earlier.

Response 12: Previous description was  about single population of two aphid species, respectively. In here, we discussed on the mixed population.

Point 13: (In line 294) Page 8, Discussion, 2nd paragraph: Was size difference a factor affecting predation of A. glycine and B. brassicae?

Response 13: There is little difference in size, 12 and 14 are much more than 6 and 8. We also changed the wording.

Point 14: (In line 344) Should not need to cite figures and tables in the Discussion.

Response 14: Deleted.

Point 15: (In line 349) Page 9, Conclusion:  Orius sauteri can be an effective…..

Response 15: We have rewritten this sentence as follows ‘Orius sauteri can be an effective potential biological control agent for P. juglandis and C. juglandicola in walnut orchards.’ 

Reviewer 3 Report

I have reviewed the manuscript “Is Orius sauteri Poppius a promising biological control agent for walnut aphids? An assessment from laboratory to field” authored by Ting et al.  Overall, I found this to be an interesting manuscript quite suitable for publication in Insects.  The authors have presented a series of experiments, lab and field, examining predation on two aphid species by a predaceous bug.  I feel that the authors should elaborate more regarding their laboratory protocol for leaf orientation and how this may have affected the aphids and predator based on later comments regarding the behavior of the aphids and the predator.

The absence of line numbers makes it more difficult to provide comments/edits to the authors.

Page 1:  ……ecosystem service substantially contributing to protecting….

The efficacy of natural enemies are commercially……

Page 2:  Last sentence of introduction is awkward – will inform the development – please re-word.

The sex of O. sauteri bugs used in experiments was not determined.

Lab experiments:  I was curious about the leaf surface orientation for this experiment:  which leaf surface (abaxial or adaxial) was oriented upward in the petri dish?  After learning, in the Discussion, that aphid preference for leaf surface is important, more detail must be provided here.  You potentially had one of the two aphid species feeding on a non-preferred leaf surface and a predator searching on a non-preferred leaf surface oriented opposite of its field preference.  Please provide more detail.

Page 5:  Table 1.  Please look at prey density for P. juglandis at 25 and 35.  The mean separation letters provide for these two numbers do not make sense.  How is 9.2 an ‘ab’ but 8.8 is an ‘a’??

Paragraph below Table 1.  You did not test/analyze the ‘rate of increase’ so comments about more/less should not be made.

Paragraph above Table 2.  You switch from saying eaten to nymphs were ‘killed’.  Was there a reason?  Being eaten vs. being killed are different and would have different implications regarding these results.

Last paragraph of the page:  …….effectively controlled populations of each aphid…….

Throughout the manuscript:  Most, if not all, instances of ‘percent’ should be ‘percentage’.

Page 6, first full paragraph.  Any statistics to support comment such as ‘control efficacy was greatest’?

Page 7, first full paragraph.  Delete ‘populations’ following C. juglandicola.

Were the populations growth rates statistically significant?

Section 3.4 seems to be a repeat of what has been stated earlier.

Last sentence of this page is not visible.

Page 8, Discussion, 2nd paragraph:  Was size difference a factor affecting predation of A. glycine and B. brassicae?

Should not need to cite figures and tables in the Discussion.

Page 9, Conclusion:  Orius sauteri can be an effective…..

Author Response

Response to Reviewer 1 Comments

We are grateful for the insightful comments on and valuable improvements on our paper. We have incorporated most of your suggestions. All changes are highlighted within the previous manuscript. Please see below with blue font for a point-by-point response to your comments and concerns.

Additionally, we added the line numbers in revised manuscript for easy reading and reviewing. We processed your comments in two ways. All of your concerns were processed in the below point to point. The other way was done word file in attachment.  

Point 1: (In line 46) Page 1:  ……ecosystem service substantially contributing to protecting….

Response 1: Changed as ‘The biological control of pests by their natural enemies is a crucial ecosystem service  substantially contributes to protecting the environment as well as maintaining biodiversity [1]. The efficacy of natural enemies are available commercially for protection of crops and trees [2]. ’ From line 46 to row 48.

Point 2: (In line 72) Page 2:  Last sentence of introduction is awkward – will inform the development – please re-word.

Response 2: Revised as ‘This assessment will inform the package of an integrated pest management program for walnut orchards.’ From line 71 to row 72.

Point 3: (In line 80) The sex of O. sauteri bugs used in experiments was not determined.

Response 3: We mention that in first paragraph in material and method: ‘The sex of O. sauteri bugs used in experiments was not identified.’ 

Point 4: (In line 94) Lab experiments:  I was curious about the leaf surface orientation for this experiment:  which leaf surface (abaxial or adaxial) was oriented upward in the petri dish?  After learning, in the Discussion, that aphid preference for leaf surface is important, more detail must be provided here.  You potentially had one of the two aphid species feeding on a non-preferred leaf surface and a predator searching on a non-preferred leaf surface oriented opposite of its field preference.  Please provide more detail.

Response 4: Sorry for poor description in the method before. We offered more explanation in here. have modified and explained in more detail, as follow: In this experiment we cut the walnut leaves equally in half which were the same size with petri dish, a half leaf with inocation P. juglandis in the adaxial surface, and another half leaf with inocation C. juglandicola in abaxial surface.

Point 5: Page 5:  Table 1.  Please look at prey density for P. juglandis at 25 and 35. The mean separation letters provide for these two numbers do not make sense.  How is 9.2 an ‘ab’ but 8.8 is an ‘a’??

Response 5: We have rewritten the mean separation letters in Table 1.

Point 6: (In line 198) Paragraph below Table 1.  You did not test/analyze the ‘rate of increase’ so comments about more/less should not be made.

Response 6: Deleted these sentences.

Point 7: (In line 72) Paragraph above Table 2.  You switch from saying eaten to nymphs were ‘killed’.  Was there a reason?  Being eaten vs. being killed are different and would have different implications regarding these results.

Response 7: We have changed ‘killed’ into eaten. From line 208.

Point 8: (In line 217) Last paragraph of the page:  …….effectively controlled populations of each aphid…….

Response 8: We have changed the wording as your comments in line 217.

Point 9: (In line 167) Throughout the manuscript: Most, if not all, instances of ‘percent’ should be ‘percentage’.

Response 9: This is a fixed terminology in ‘ percent reduction of cumulative insect-days ’. See the reference in 23.

Point 10: (In line 225) Page 6, first full paragraph. Any statistics to support comment such as ‘control efficacy was greatest’?

Response 10: We have rewritten this paragraph, as follows: ‘The populations of P. juglandis, as well as C. juglandicola, declined after 15 days if treated with predators. The mean control efficacy of P. juglandis were 79%±8.7% with 1 predator and 92%±4.3% with 2 predators in laboratory while in the field the means were 54%±8.7% and 84%±6.1% with 1 or 2 predators). In case of C. juglandicola the mean control efficacy were 63%±13.2% and 95%±4.6% with 1 or 2 predators in laboratory but in the field the mean were 76%±6.1% and 90%±2.3% with 1 or 2 predators. The mean control efficacy of O. sauteri were highest by using these two predator densities.’

Point 11: (In line 225) Page 7, first full paragraph.  Delete ‘populations’ following C. juglandicola. Were the populations growth rates statistically significant?

Response 11: We have deleted ‘populations’ following C. juglandicola in full paragraph, from line 217 to 223. And, the populations growth rates were statistically significant in the presence and absence of different initial densities O. sauteri bugs. We rewrite this paragraph as follows ‘The mean population growth rates(PGR) of P. juglandis were 0.1±0.01, -0.06±0.03 and -0.12±0.03 with 0, 1 or 2 predators in laboratory while in the field the mean were 0.15±0.01, 0.08±0.02 and -0.03±0.03 with 0, 1 or 2 predators. In case of C. juglandicola, mean were 0.22±0.01, 0.07±0.05 and -0.11±0.03 with 0, 1 or 2 predators in laboratory and in the field the mean were 0.16±0.01, 0.05±0.01 and -0.02±0.02 with 0, 1 or 2 predators. The mean population growth rates(PGR) were significantly different in the presence and absence of different initial densities O. sauteri bugs (H=19.1, df=2, P<0.05 for P. juglandis in laboratory; H=18.1, df=2, P<0.05 for C. juglandicola in laboratory; H=18.2, df=2, P<0.05 for P. juglandis in field; H=21.4, df=2, P<0.05 for P. juglandis in field). Aphid populations treated with two adult O. sauteri had lower PGRs than those treated with one. The PGR of P. juglandis populations (-0.12±0.03) was less than C. juglandicola (-0.11±0.03) in laboratory experiments. Conversely, the PGR of P. juglandis populations (-0.03±0.02) was greater than C. juglandicola (-0.02±0.02) treated with two predatory bugs in field experiments (Fig. 2).’ 

Point 12: Section 3.4 seems to be a repeat of what has been stated earlier.

Response 12: Previous description was  about single population of two aphid species, respectively. In here, we discussed on the mixed population.

Point 13: (In line 294) Page 8, Discussion, 2nd paragraph: Was size difference a factor affecting predation of A. glycine and B. brassicae?

Response 13: There is little difference in size, 12 and 14 are much more than 6 and 8. We also changed the wording.

Point 14: (In line 344) Should not need to cite figures and tables in the Discussion.

Response 14: Deleted.

Point 15: (In line 349) Page 9, Conclusion:  Orius sauteri can be an effective…..

Response 15: We have rewritten this sentence as follows ‘Orius sauteri can be an effective potential biological control agent for P. juglandis and C. juglandicola in walnut orchards.’ 

Round 2

Reviewer 1 Report

The authors have responded well to each comment. There is one tiny grammatical error in response 3. This actually comes from me. 

'The two aphid species colonize walnut trees and the same time with the relatively large aphid..' should read 'The two aphid species colonize walnut trees at the same time with the relatively large aphid..'

Just something small, otherwise looks good.

Author Response

Response to Reviewer  Comments

We appreciate the time and effort that you dedicated to providing feedback on our manuscript and are grateful for the insightful comments on and valuable improvements to our paper.

At the sametime we also rephrased the sentence that you mentioned, and upload the file in Word.
